# Advances in Genomic Data and Biomarkers: Revolutionizing NSCLC Diagnosis and Treatment

**DOI:** 10.3390/cancers15133474

**Published:** 2023-07-03

**Authors:** Juan Carlos Restrepo, Diana Dueñas, Zuray Corredor, Yamil Liscano

**Affiliations:** 1Grupo de Investigación en Salud Integral (GISI), Departamento Facultad de Salud, Universidad Santiago de Cali, Cali 760035, Colombia; juan.restrepo16@usc.edu.co (J.C.R.); diana.duenas00@usc.edu.co (D.D.); 2Grupo de Investigaciones en Odontología (GIOD), Facultad de Odontología, Universidad Cooperativa de Colombia, Pasto 520002, Colombia; 3Facultad de Salud, Departamento de Ciencias Básicas, Universidad Libre, Cali 760026, Colombia; zurayf.corredorm@unilibre.edu.co

**Keywords:** biomarkers, miRNA, bioinformatics, non-small cell lung cancer, bioprospecting, machine learning, artificial intelligence

## Abstract

**Simple Summary:**

In the field of non-small cell lung cancer (NSCLC), there have been significant advancements in genomic data and bioinformatics tools, which have improved early diagnosis, treatment, and follow-up using biomarkers. Biomarkers provide measurable indicators of disease characteristics and help tailor treatment strategies in precision medicine. The integration of big data and artificial intelligence (AI) further enhances personalized medicine through advanced biomarker analysis. However, challenges exist in terms of limited evidence on the impact of new biomarkers on mortality and treatment efficacy, as well as data analysis and the adoption of precision medicine in clinical practice. Despite these obstacles, the integration of biomarkers into precision medicine has shown promise in improving patient outcomes in NSCLC. Continued research and advancements in biomarker discovery, utilization, and evidence generation are needed to overcome these challenges and further enhance precision medicine’s effectiveness in NSCLC.

**Abstract:**

Non-small cell lung cancer (NSCLC) is a significant public health concern with high mortality rates. Recent advancements in genomic data, bioinformatics tools, and the utilization of biomarkers have improved the possibilities for early diagnosis, effective treatment, and follow-up in NSCLC. Biomarkers play a crucial role in precision medicine by providing measurable indicators of disease characteristics, enabling tailored treatment strategies. The integration of big data and artificial intelligence (AI) further enhances the potential for personalized medicine through advanced biomarker analysis. However, challenges remain in the impact of new biomarkers on mortality and treatment efficacy due to limited evidence. Data analysis, interpretation, and the adoption of precision medicine approaches in clinical practice pose additional challenges and emphasize the integration of biomarkers with advanced technologies such as genomic data analysis and artificial intelligence (AI), which enhance the potential of precision medicine in NSCLC. Despite these obstacles, the integration of biomarkers into precision medicine has shown promising results in NSCLC, improving patient outcomes and enabling targeted therapies. Continued research and advancements in biomarker discovery, utilization, and evidence generation are necessary to overcome these challenges and further enhance the efficacy of precision medicine. Addressing these obstacles will contribute to the continued improvement of patient outcomes in non-small cell lung cancer.

## 1. Introduction

Non-small cell lung cancer (NSCLC) is a serious public health problem, accounting for 84% of diagnosed lung cancers. In 2020, there were 2,206,771 (11.4%) new diagnoses of lung cancer in both sexes and all ages, second only to breast cancer. However, it had the highest mortality rate with 1.8 million deaths in 2020 [1], mainly due to late diagnosis. With the increase in genomic data, the possibilities for early diagnosis, effective and safe treatment, and follow-up are enhanced by increasing the likelihood of prediction through biomarkers. This is achieved by using bioinformatics tools and automated machine learning algorithms to analyze large amounts of data, including clinical data obtained from harmonized medical records, radiological images, and pathological studies (tissue biopsies), as well as genomic data.

The rapid growth of genomic data in cancer and the development of bioinformatics analysis methods have led to the identification of tumor biomarkers that facilitate early detection, treatment, and prognosis, reducing mortality rates in some types of cancer [2]. Public data sources such as the Gene Expression Omnibus (GEO) and The Cancer Genome Atlas (TCGA) provide opportunities to explore tumor genesis and progression, as well as the identification of new biomarkers for diagnosis, prognosis, and treatment response [3]. Biomarkers, combined with clinical data, have prognostic value and can predict outcomes, guiding specific treatments [4]. To structure and perform classification and prediction tasks with this big data, different machine learning methods have evolved according to the needs and magnitude of the data [5]. In a multi-omic conceptual framework, integrating component analysis, non-parametric combinations, and exploratory analysis contribute to consistent information and enable the application of classification and prediction algorithms such as decision trees, random forests, support vector machines, linear and logistic regression, and deep learning models. Recently, dimensionality reduction techniques such as t-distributed stochastic neighbor embedding (t-SNE), multidimensional scaling (MDS), and uniform manifold approximation and projection (UMAP) have been applied based on data structure and acquisition methods [6].

In the case of non-small cell lung cancer, the impact of a new biomarker on mortality and the efficacy of specific treatment is a challenge due to the limited evidence available. Therefore, the discovery and application of relevant biomarkers, both new and existing, will lead the future of precision medicine in NSCLC [7,8].

## 2. Non-Small Cell Lung Cancer

NSCLC is the most common type of lung cancer, accounting for approximately 85% of all cases [9], and it is the leading cause of cancer-related deaths worldwide, ranking first in men and second in women, with nearly 1.8 million deaths per year [10]. It is a complex and diverse disease that arises from the abnormal growth of cells in the lung tissue, primarily caused by smoking. Early detection and increased utilization of diagnostic tools such as PET scans and the discovery of biomarkers are crucial for improving patient outcomes and reducing lung cancer mortality rates [11,12].

NSCLC encompasses various subtypes, including adenocarcinoma, squamous cell carcinoma, and large cell carcinoma. This type of cancer is often diagnosed at advanced stages, which poses challenges for effective treatment. Adenocarcinoma is the most common type of lung cancer, accounting for approximately 40% of cases. It develops from small airway epithelial cells known as type II alveolar cells, which secrete mucus and other substances. Adenocarcinoma affects both smokers and non-smokers and occurs in men and women of all ages. It tends to occur in the outer regions of the lungs, possibly due to cigarette filters preventing large particles from entering the lungs. Adenocarcinoma generally grows slower and has a higher chance of being detected before it spreads beyond the lungs compared to other types of lung cancer [10,11].

Large cell carcinoma (undifferentiated) represents 5–10% of lung cancers. This type of carcinoma lacks evidence of squamous or glandular maturation and is often diagnosed by excluding other possibilities. Large cell carcinoma typically originates in the central part of the lungs and can spread to nearby lymph nodes, the chest wall, and distant organs. This type of carcinoma is strongly associated with tobacco consumption [9,10].

### 2.1. Risk Factors and Epidemiology

Some factors that indicate a negative outlook include being male, having impaired functional abilities, and being over the age of 70. The use of radiological imaging does not effectively reduce mortality because of the aggressive nature of the disease. The most crucial steps to reduce mortality are quitting smoking and prevention, given that smoking is the primary risk factor for lung cancer. Non-smokers exposed to secondhand tobacco smoke also face an increased risk of developing lung cancer, and living with a smoker can raise the chances by 20–30% [9,10,13].

Tobacco consumption has been identified as the cause of 90% of all lung cancer cases. Current smokers with a history of 40 pack-years have twenty times higher chances of developing lung cancer than a nonsmoker. This risk can increase when additional environmental or lifestyle exposures are combined with tobacco consumption, such as asbestos exposure. It is believed that adenocarcinoma, in particular, originated from the invention of filtered cigarettes in the 1960s, although this has not been proven [14,15].

Another risk factor is radon, a natural carcinogen present in uranium deposits in basements, which is associated with lung cancer and is estimated to have caused approximately 21,000 deaths from this disease in the United States. Occupational exposure to substances such as asbestos, arsenic, beryllium, and other chemicals also increases the risk of lung cancer. Air pollution, especially in areas with heavy traffic and high concentration of pollutants, including polycyclic aromatic hydrocarbons, is identified as a risk factor for lung cancer, with an 8% increase in overall mortality risk from this disease. Personal or family history of lung cancer are also risk factors, and certain genes and chromosomes associated with a higher risk of developing the disease, such as the TP53 gene and a marker on chromosome 15, have been identified [9,10,13].

On a global level, lung cancer is the leading cause of cancer-related death in men and the second most common in women. There is significant variation in the incidence of lung cancer across different populations, primarily driven by the prevalence of tobacco consumption in various countries. The incidence of lung cancer is directly related to the rise or decline in smoking rates among different populations. For instance, it is projected that age-adjusted mortality rates in the United States will decrease by 79% between 2015 and 2065 due to declining tobacco use rates and anti-smoking campaigns. Incidence and mortality rates of lung cancer are higher in developed countries. In contrast, lung cancer rates in underdeveloped geographic areas, such as Central/South America and much of Africa, are lower [14,15].

However, many developing countries lack a centralized reporting system, leading to underreporting of lung cancer cases, making it challenging to determine the true incidence of the disease. The World Health Organization (WHO) estimates that global lung cancer mortality rates will continue to rise, primarily due to the increasing global tobacco consumption, particularly in Asia. In the United States, lung cancer incidence and mortality rates have been declining in men, while initially increasing in women until around the year 2000, and have since stabilized. Due to the shifting incidence of lung cancer in women, mortality rates have decreased in women over a decade after declining in men [14,15].

According to data from the Global Cancer Observatory (GLOBOCAN), lung cancer is the second most common cancer worldwide, following female breast cancer. In terms of mortality, lung cancer is the leading cause of cancer-related death. In South America, the incidence and mortality rates of lung cancer vary among countries. In 2020, the incidence of lung cancer in South America was 17.8 cases per 100,000 population in men and 10.3 in women. In Colombia, lung cancer ranked fifth in incidence in 2020, following breast, prostate, colorectal, and stomach cancer. In terms of mortality, lung cancer is the second leading cause of cancer-related death in Colombia, with approximately 9.2 deaths per 100,000 population each year [16,17]. Adenocarcinoma is the most common histological variant in lung cancer patients in Colombia, followed by squamous cell carcinoma. The majority of patients present at advanced stages of the disease. Mutations in the Epidermal Growth Factor Receptor (EGFR) gene are detected in a significant proportion of patients, mainly in non-smoking women. ALK rearrangement is also found in a small proportion of cases. It is necessary to implement lung cancer prevention and control strategies to reduce its incidence and mortality in Colombia [16,17].

Lung cancer has a high social and economic impact globally, causing a significant number of disability-adjusted life years lost. In Latin America, lung cancer treatment is costly and limited in terms of the adoption of new technologies and medications. In Colombia, lung cancer mortality rates have shown a decreasing trend since 2005, both in men and women, across all age groups [16,17]. The decline in mortality rates in men aged 35 to 64 was observed in 1985, while in women of the same age group, it was observed in 1991. However, in the group of individuals older than 64, the decline in mortality rates began to be observed in the early 21st century. The study by Giraldo-Osorio et al., from 2022 is the first in over 30 years to determine the trend in lung cancer mortality at the national level in Colombia and uses joinpoint regression analysis. It highlights the increase in lung cancer mortality rates in women over 64 years old until the early 21st century and in women under 65 years old until 1991 [16]. This could be related to exposure to solid fuel smoke, especially in indigenous women over 30 years old living in rural areas. The use of solid fuels for cooking has decreased in Colombia, which may have contributed to the decline in lung cancer mortality rates in women. The decrease in tobacco consumption prevalence in Colombia may explain the reduction in mortality rates, especially in men [16]. Over the decades, there has been a decrease in tobacco consumption prevalence in the general population, although men have higher rates than women. Laws and projects for tobacco control have been implemented, contributing to the reduction in consumption. It is important for Colombia to continue efforts in primary prevention to prevent the initiation of tobacco use and in secondary prevention to assist people in quitting smoking [16].

### 2.2. Pathophysiology, Histology, and Classification

Oncogene mutations play a significant role in NSCLC and can be effectively targeted with specific drugs. Familial clustering of lung cancer has provided evidence for a hereditary component in disease development. Carriers of TP53 germline sequence variations have an increased risk of lung cancer, particularly if they are smokers. A genome-wide linkage study has identified a susceptibility locus at 6q23-25p that influences lung cancer risk [13]. The most common mutation occurs in the EGFR gene, affecting 10–30% of patients. This mutation is associated with downstream signaling pathways, including MAPK/ERK, PI3K/AKT, and Bax/Bcl-2. The presence of the EGFR T790M sequence variation has been identified in families with multiple cases of NSCLC. Additionally, it is linked to acquired resistance to EGFR tyrosine kinase inhibitors (TKIs) [18]. Osteopontin (OPN) and the PI3K/AKT/mTOR pathway also contribute to resistance against EGFR TKIs. Gene rearrangements in the RET gene are detected in approximately 1% to 2% of patients with NSCLC and affect signaling pathways such as PI3K/AKT, JAK-STAT, and RAS/MAPK. These pathways play a role in cell proliferation, invasion, and migration. Mutations in the MET gene lead to abnormal expression of the MET axis, promoting the migration and invasion of tumor cells, as well as resistance to inhibitors targeting EGFR and VEGFR. Rearrangements in the Anaplastic Lymphoma Kinase (ALK) gene occur in 5% to 6% of NSCLC patients, resulting in increased expression of ALK, which triggers epithelial-mesenchymal transition (EMT) and enhances migration and invasion. Resistance to ALK inhibitors is associated with specific mutations such as F1174L and involves the STAT3/ZEB1 signaling pathway [18]. Human epidermal growth factor receptor 2 (HER2) gene mutations, particularly in exon 20, are found in 2–4% of NSCLC patients, especially in women, and are associated with brain metastases. Activation of HER2 triggers downstream signaling through MEK/ERK and PI3K/AKT pathways, promoting lung cancer cell migration and proliferation. BRAF mutations, particularly V600E, confer resistance to BRAF inhibitors but respond better to combined BRAF and MEK inhibitors. ROS1 gene rearrangements are present in 1–2% of NSCLC patients, with CD74-ROS1 being the most common fusion. Activation of the ROS1 gene initiates signaling through the PI3K/AKT/mTOR, JAK/STAT, and MAPK/ERK pathways, facilitating the proliferation and invasion of tumor cells. In NSCLC patients, KRAS mutations are identified in around 13% of cases, predominantly among smokers, and are linked to drug resistance and unfavorable outcomes. Various KRAS mutations activate specific downstream signaling pathways. Gene fusions involving the Neurotrophic tropomyosin receptor kinase (NTRK) gene occur in less than 1% of NSCLC patients and affect the MEK/ERK and PI3K/AKT pathways, which play a role in cell proliferation, migration, and chemotherapy-induced apoptosis. Early-stage NSCLC patients with NTRK fusions exhibit a significant response rate to TKIs [12,18].

The genomic characteristics of lung cancer differ significantly between smokers and never-smokers. Smokers display a higher frequency of mutations, including non-actionable mutations in genes such as KRAS and TP53, characterized by specific nucleotide changes. On the other hand, never-smokers more commonly exhibit specific actionable gene alterations such as activating EGFR mutations and ROS1 and ALK translocations. These genetic events that drive tumor development also impact the composition of the tumor microenvironment (TME). NSCLC, in particular, demonstrates a high number of somatic tumor mutations, especially in smokers, and metastatic tumors tend to have more mutations than primary lung lesions. Certain mutations result in the formation of neoantigens, which can be recognized by immune cells infiltrating the tumor. An increased burden of these neoantigens is associated with an inflamed TME, enriched with activated immune cells and the expression of immune-related proteins. Additionally, defects in DNA mismatch repair and microsatellite instability contribute to a high tumor mutation burden and favorable responses to immune checkpoint inhibitors (ICBs). Furthermore, genetic alterations can influence the cellular composition and functions of the TME. For instance, the inactivation of the STK11 tumor suppressor gene in KRAS-mutated lung adenocarcinoma shifts the TME towards immunosuppressive neutrophils and reduced expression of PD-L1. Further investigations are required to fully understand the intricate relationship between genotypes of NSCLC and the cellular makeup of the TME [12].

Histology plays a crucial role in cancer diagnosis, allowing for the distinction of different subtypes of lung tumors. In adenocarcinoma, evidence of neoplastic gland formation, expression of markers such as TTF-1 and napsin, or the presence of intracytoplasmic mucin is required. Squamous cell carcinoma is diagnosed by detecting keratin production and intercellular desmosomes. Immunohistochemistry (IHC) enables the identification of specific markers such as p40, p63, CK5, or desmoglein in squamous cell carcinoma. On the other hand, large cell carcinoma is a diagnosis of exclusion as it can exhibit features of squamous, glandular, or neuroendocrine differentiation. In cases of poorly differentiated carcinoma, it will be classified as large cell carcinoma only if specific markers indicating another subtype of lung cancer are not found [15].

There are two main types of lung cancer: NSCLC, which accounts for 85% of cases, and small cell lung cancer (SCLC), which makes up 15% of cases. NSCLC is further classified into three main types: adenocarcinoma, squamous cell carcinoma, and large cell carcinoma. Adenocarcinoma is the most common subtype and arises from alveolar cells in the smaller airway epithelium. Squamous cell carcinomas arise from cells in the airway epithelium, and large cell cancers are typically poorly differentiated and composed of large cells. Immunohistochemical markers are used to identify these subtypes (Figure 1) [14,15].

### 2.3. Resistance to Medications and Immunotherapy in NSCLC: Mechanisms and Therapeutic Strategies

Drug resistance is one of the main causes of therapeutic failure in NSCLC, leading to tumor recurrence and disease progression. Resistance in NSCLC is a significant clinical problem as it diminishes treatment efficacy and may contribute to disease progression and the development of metastasis. The clinical relevance of resistance in NSCLC lies in the need to develop more effective therapeutic approaches to overcome it. Research is focused on identifying the mechanisms responsible for resistance and finding strategies to counteract them. This includes the development of therapies specifically targeting genetic mutations present in tumor cells, as well as the use of combinations of different treatments to address resistance more effectively [19,20].

NSCLC is a heterogeneous disease that exhibits various genetic alterations. Some of the most relevant therapeutic targets in NSCLC include EGFR, ALK, ROS1, BRAF, MET, RET, and KRAS. The intrinsic mechanisms of cellular resistance in this disease encompass changes in drug transporter expression and activation of pro-survival and anti-apoptotic pathways, along with non-intrinsic influences from the tumor microenvironment [19]. TKIs have proven to be effective in the treatment of NSCLC in patients with activating mutations in these genes. For instance, EGFR TKIs such as erlotinib and gefitinib have shown promising results in patients with EGFR mutations. Similarly, ALK TKIs such as crizotinib and alectinib have demonstrated activity in patients with ALK fusions. Additionally, targeted drugs have been developed for ROS1, BRAF, MET, RET, and KRAS, thereby expanding treatment options for NSCLC patients harboring these genetic alterations [20].

The genetic alterations and tumor heterogeneity in NSCLC are driven by subpopulations of tumor cells called cancer stem cells (CSCs), which possess tumor-initiating capabilities, high self-renewal capacity, and the ability to differentiate into multiple lineages. CSCs have been identified in NSCLC and have been associated with resistance to chemotherapy, radiation therapy, and immunotherapy [19,21].

Resistance to immunotherapy can manifest at different levels of dysfunction and be related to specific tumor phenotypes. One mechanism of resistance involves the absence of specific T lymphocytes in the tumor microenvironment or their inability to express the necessary T cell receptor (TCR) to recognize the tumor antigen. Another mechanism is related to the lack of infiltration of activated T lymphocytes into the tumor tissue due to the presence of substances such as VEGF that promote the formation of barriers for immune cells [21].

In CPCNP, resistance to immunotherapy can also arise, such as in the case of immune checkpoint inhibitors. The tumor microenvironment and interactions between cancer cells and immune cells play a crucial role in determining the response to immunotherapy. Tumors with a high population of CSCs have been found to exhibit immunosuppressive characteristics and evade immune surveillance, leading to resistance to immunotherapy [19]. CSCs can modulate immune responses through various mechanisms, including the expression of immune inhibitory molecules, secretion of immunosuppressive factors, and induction of dysfunction in immune cells [19]. On the other hand, resistance to EGFR inhibitors is common in patients with lung adenocarcinoma. Various mechanisms of resistance have been identified, including EGFR mutations, oncogenic changes, alterations in apoptosis, and epithelial-mesenchymal transition. EGFR inhibition can lead to the enrichment of the cancer stem cell subpopulation in non-small cell lung cancer, whether mutated or non-mutated in EGFR, through a NOTCH3-dependent process. Additionally, this inhibition can increase cell death in non-stem cells and enhance the formation of pulmonary spheres [19]. Inhibiting NOTCH1 and HES1 can reverse resistance to gefitinib by increasing apoptosis. Dual inhibition of EGFR and NOTCH2/3 reduces EGFR and radiation-induced cancer stem cell subpopulations, as well as the expression of DNA repair genes. Similarly, resistance to ALK inhibitors in NSCLC may be mediated by mechanisms associated with NOTCH signaling. Erlotinib faces resistance due to mutations such as T790M, D761Y, L747S, and T854A in the EGFR gene, as well as EGFR amplification, PIK3CA gene mutations, and MET gene amplification. The MET gene encodes a receptor tyrosine kinase (RTK) protein that is activated by binding with the HGF ligand, triggering a signaling cascade that promotes cellular progression and proliferation in various malignant tumors. This amplification can occur early in tumorigenesis or as a mechanism of acquired resistance to tyrosine kinase inhibitors. However, specific MET inhibitors and multi-kinase inhibitors have been developed, showing promising results in treating tumors with MET gene amplification, thereby improving clinical outcomes [22]. Studies have shown that both overexpression and amplification of MET are associated with poor prognosis in patients with NSCLC, and MET amplification appears to be an independent marker of poor outcome after surgical resection of NSCLC. In a series of 687 Asian patients with resected NSCLC, MET alterations were unfavorable prognostic factors for overall survival [23].

Regarding ALK inhibitors, mutations such as G1269A, C1156Y, I1171T/N/S, S1206C, E1210K, L1152P/R, V1180L, G1128A, F1174V, and L1196M in the ALK kinase domain, along with bypass pathway activation, KIT amplification, and MAPK signaling, contribute to resistance. ROS1 inhibitors face resistance due to point mutations in the ROS1 kinase domain, such as D2033N, G2032, L2026M, L2155S, and S1986F/Y, as well as the involvement of other receptors and effectors in the MAPK pathway. Lastly, BRAF inhibitors are affected by mutations such as BRAF(V600E), BRAF(D594G), and BRAF(G469A/V), as well as bypass pathway activation and restoration of MAPK signaling [20].

Regarding anti-angiogenic therapy targeting VEGF and DLL4-NOTCH, resistance can arise due to the activation of alternative pro-angiogenic signals. In KRAS-driven tumors, NOTCH inhibition can improve treatment sensitivity and suppress RAS and PI3K signaling pathways [19]. The KRAS gene is frequently mutated in NSCLC, and although it has long been considered elusive, specific inhibitors such as KRASG12C inhibitors have been developed. However, resistance mechanisms have also been identified, both KRAS-dependent and KRAS-independent, involving acquired alterations in KRAS, activation of alternative signaling pathways, and histological transformation [24].

In the field of therapy based on immune checkpoint inhibitors (ICIs), which target PD-1 and PD-L1, a revolution has been observed in the treatment of NSCLC. However, not all patients respond to this therapy, and different resistance mechanisms have been identified. Primary resistance is related to the inability of the immune system to activate an adequate response against cancer cells, influenced by intrinsic and extrinsic factors to the tumor. Acquired resistance occurs when tumor cells adapt to the immune system and evade its response. Furthermore, cancer progression can also be observed after discontinuation of treatment, which may involve elements of both primary and acquired resistance [20,24]. Intrinsic resistance to immunotherapy can be attributed to various genetic aberrations in oncogenes and tumor suppressor genes, which affect the immune response and alter cytokine profiles and immune cell composition. For example, certain genetic alterations, such as RET gene rearrangements and HER2 gene mutations, are associated with low PD-L1 expression, while activating mutations in the EGFR gene and ALK gene rearrangements are associated with high PD-L1 expression but a limited response to ICIs. Other genetic factors, such as KRAS and STK11/LKB1 gene mutations, have been linked to both sensitivity and resistance to ICIs. The STK11/AMPK axis is an intracellular signaling pathway that plays a crucial role in the regulation of metabolism and energy homeostasis in cells. STK11 (also known as LKB1) is a protein kinase that acts as a key regulator in this pathway, while AMPK (AMP-activated protein kinase) is its main substrate. When cellular AMP levels increase, as occurs during situations of energy stress or intense physical exercise, AMPK is activated through phosphorylation by STK11. Once activated, AMPK triggers a series of adaptive responses to restore energy balance. This includes the inhibition of protein and lipid synthesis, as well as the activation of catabolic pathways that generate ATP, such as glycolysis and fatty acid oxidation [25]. LKB1 deficiency, observed in tumors such as lung adenocarcinomas, is associated with lower overall survival and the formation of aggressive tumors with metastatic characteristics. Additionally, the absence of LKB1 promotes EMT, affects cell polarity, and creates an immunosuppressive microenvironment around tumor cells. In the treatment of advanced NSCLC, STK11 mutations are associated with increased resistance to radiotherapy and anti-PD-L1 therapy. Loss of LKB1 due to mutations in STK11 contributes to cancer progression and resistance to certain therapies, making it a potential target for the development of more effective lung cancer treatments [25].

Although ICI has significantly improved overall survival in patients with certain types of cancer, it still has limitations and challenges. One of the main obstacles is acquired resistance to therapy, which has been associated with decreased MHC-I in the presentation of tumor antigens to T cells. However, several therapeutic strategies are being explored to increase MHC-I expression and improve the effectiveness of immunotherapy. These approaches include activation of the interferon-gamma signaling pathway, stimulation of the NF-kB signaling pathway, use of chemotherapeutic agents, and inhibition of autophagy. Although further research is still needed, these strategies could provide new opportunities to enhance the immunotherapeutic response in cancer patients [26].

On the other hand, mutations in the BRAF gene have been reported in approximately 4% of NSCLC cases, being more common in non-small cell lung adenocarcinoma. The BRAF gene encodes a kinase protein that plays a crucial role in the regulation of cell growth and proliferation. Mutations in the BRAF gene, especially the V600E mutation, have been identified in various types of cancer, including melanoma and NSCLC. Activation of the BRAF gene in NSCLC has been associated with uncontrolled cell proliferation and resistance to apoptosis, contributing to tumor growth. Although BRAF mutant inhibition has proven effective in treating melanoma, its efficacy in NSCLC is limited due to the development of resistance. Resistance to BRAF inhibition in NSCLC can be primary or acquired. Primary resistance is due to intrinsic mechanisms that limit the effectiveness of BRAF inhibition. On the other hand, acquired resistance develops during treatment and can be caused by the activation of alternative signaling pathways, such as the EGFR pathway or the PI3K/AKT/mTOR signaling pathway. Several strategies have been proposed to overcome resistance to BRAF inhibition in NSCLC. These include the combination of BRAF inhibitors with MEK inhibitors to block multiple signaling pathways, the use of EGFR inhibitors in combination with BRAF inhibitors, and the combination of targeted therapies with immunotherapy to enhance the immune response against tumor cells. Although significant progress has been made in the treatment of resistance to BRAF inhibition in NSCLC, further research is still needed to fully understand the resistance mechanisms and develop more effective therapeutic strategies [27,28].

Another resistance found in NSCLC is EMT. EMT is a biological process that can drive metastasis in various types of cancer, including NSCLC. EMT has been associated with immune exclusion and resistance to immunotherapy in melanoma and other types of cancer. EMT has been found to be associated with the exclusion of critical immune cells in the immune response against cancer, such as CD4 and CD8 T cells. Additionally, certain immunosuppressive cytokines and immune checkpoint molecules have been identified to be associated with EMT in NSCLC, suggesting potential mechanisms of resistance to immunotherapy. It is also important to note that EMT is considered a key factor in the metastasis of lung cancer. EMT involves the transformation of epithelial cells into mesenchymal cells, promoting cancer progression and transforming cells into cancer stem cells (CSCs). Various signaling pathways and transcription factors, such as Snail, TWIST, ZEB, and FOXC2, play a significant role in regulating EMT in lung cancer. These factors contribute to the loss of cell adhesion, activation of proteolysis, and increased cell motility, leading to tumor metastasis. EMT is induced by signaling pathways such as TGF-β, which activates the SMAD, PI3K-AKT, and MAPK pathways. Overall, these findings highlight the importance of understanding how EMT affects the tumor microenvironment and immune response in NSCLC to develop more effective therapeutic strategies [29,30].

In conclusion, it is crucial to continue research in NSCLC to identify new markers of resistance and develop effective therapeutic strategies. The identification of these markers and the development of appropriate treatment approaches are essential for improving patient outcomes and overcoming the challenges posed by this disease. 

### 2.4. Diagnosis and Treatment

Early detection of lung cancer through screening is of utmost importance as it can significantly improve clinical outcomes. During the initial stages, most patients with lung cancer do not exhibit noticeable symptoms, apart from coughing. This lack of symptoms can lead to missed opportunities for early diagnosis and treatment. Diagnostic approaches encompass imaging tests, biopsies, and biomarker tests. Imaging tests, including computed tomography (CT), positron emission tomography (PET), and magnetic resonance imaging (MRI), are vital in diagnosing lung cancer. CT screening, in particular, has shown a 20% reduction in lung cancer mortality and a decrease in overall mortality rates [14,18]. Advancements in diagnostic technologies play a critical role in saving lives and extending the lifespan of patients. Among these technologies, computed tomography (CT) is widely used as a diagnostic method, enabling the determination of tumor size and the identification of nodules in individuals with lung cancer. In order to detect lung cancer at an early stage, the US Preventive Services Task Force recommends annual low-dose CT screening for individuals at high risk. Furthermore, the NELSON trial demonstrated a noteworthy 26% decrease in lung cancer mortality among high-risk individuals who underwent low-dose CT screening [11,13,14]. It can also detect metastases, especially in mediastinal lymph nodes. However, biopsy is still necessary to determine if the nodules are benign or malignant. which can be tissue- or liquid-based. Tissue biopsy is invasive and considered the gold standard in clinical practice. It can determine the different histological types of lung cancer and detect mutations. However, it can have complications. Liquid biopsy is less invasive and utilizes peripheral blood samples. It can detect circulating tumor DNA, circulating tumor cells, exosomes, and other molecules such as miRNA and circRNA. Although it is more sensitive and practical than tissue biopsy, its application is limited [14,18].

PET, unlike CT, exhibits greater sensitivity and specificity due to its utilization of abnormal glucose markers. This imaging technique is capable of identifying malignant lesions characterized by abnormal glucose metabolism, determining their benign or malignant nature, and distinguishing between various types and stages of lung cancer. On the other hand, MRI is employed to detect metastases in the brain and bones of individuals with NSCLC. Through advancements in high-performance gradient systems, phased-array receiver coils, and optimized imaging sequences, MRI can now detect even small nodules in lung tissues, reaching a size as minute as 3 mm [31].

When it comes to biomarker tests, it is advisable to perform molecular tests on patients diagnosed with NSCLC to assess potential mutations. Several techniques are employed for this purpose, including polymerase chain reaction (PCR), fluorescence in situ hybridization (FISH), immunohistochemistry (IHC), and next-generation sequencing (NGS). These tests play a significant role in identifying various mutations associated with lung cancer and are instrumental in the development of targeted therapies. Precise determination of the TNM stage of lung cancer is essential for selecting the most appropriate treatment approach [14,18,32].

In the age of precision medicine, immunohistochemistry (IHC) plays a vital role in categorizing tumors into distinct subtypes and evaluating biomarkers to guide effective therapeutic decision-making. IHC offers several advantages over other techniques, such as its widespread availability, technical simplicity, cost-effectiveness, and rapid turnaround time. It has the potential to serve as a practical screening tool for identifying targetable genetic alterations and assessing biomarkers for molecular-targeted therapies. Furthermore, IHC provides valuable information about the cellular distribution of proteins within tumor tissues, enabling a more comprehensive understanding of the disease (see Table 1) [32]. Despite the benefits, challenges to lung cancer screening adoption include insurance coverage, false-positive results, radiation exposure concerns, patient distress, and the risk of overdiagnosis [14,18].

## 3. Precision Medicine and Biomarkers

### 3.1. The Impact of Precision Medicine

Precision medicine is a groundbreaking discipline that is transforming the way medicine is practiced. It emerges as a response to the growing understanding that each patient is unique, with genetic and molecular characteristics that can influence their susceptibility to diseases, response to treatments, and prognosis. Precision medicine is important today because it allows for a more individualized and personalized approach to the diagnosis, treatment, and prevention of diseases. Unlike traditional approaches that rely on general protocols, precision medicine takes into account the unique characteristics of each individual, such as their genetics, lifestyle, environment, and other factors, to make more informed medical decisions. Precision medicine relies on the gathering and examination of diverse data sets, such as genetic, molecular, clinical, and lifestyle information, to gain a deeper understanding of the fundamental biology of a disease and to make well-informed choices regarding diagnosis and treatment options [33,34].

Precision medicine has also driven significant advances in the field of oncology. Through genomic and molecular analysis of tumors, specific alterations in a patient’s DNA can be identified, allowing for the selection of therapies targeted at these alterations. This has revolutionized cancer treatment as we can now use specific drugs that act on the altered molecular signaling pathways in cancer cells, improving treatment efficacy and reducing side effects. Additionally, precision medicine also plays a crucial role in disease prevention. By better understanding the genetic and environmental risk factors of each individual, proactive measures can be taken to prevent the onset of diseases or detect them at early stages when they are more treatable. This enables personalized preventive interventions such as lifestyle changes, more frequent medical screenings, or even the use of preventive medications in selected cases [35,36].

A fundamental component of precision medicine is the utilization of biomarkers, which play a crucial role in identifying and understanding specific characteristics in a patient. Biomarkers are measurable and objective indicators that can encompass a wide range of aspects, from genetic mutations and gene expression profiles to protein levels and distinctive clinical features. These biomarkers offer valuable insights into the underlying biological aspects of a disease and enable healthcare professionals to obtain a more comprehensive and detailed understanding of a patient’s medical condition. For example, specific genetic mutations may indicate the presence of an inherited disease or susceptibility to certain types of cancer. Gene expression profiles can reveal molecular patterns associated with a positive or negative response to a particular treatment. Protein levels in the body can indicate the presence or progression of a disease. By analyzing these biomarkers, physicians can tailor treatment more precisely to each patient. This means that therapeutic approaches can be customized based on the specific molecular and genetic characteristics of each individual, increasing the likelihood of a successful outcome [37,38,39].

Big data and artificial intelligence (AI) are becoming increasingly crucial in the analysis and interpretation of biomarkers within the realm of precision medicine. With the advancement of technology, there has been an explosion in the generation and collection of large-scale health data, leading to the formation of large datasets, known as big data. These datasets include genetic information, clinical records, medical images, laboratory test results, and much more. AI, in combination with big data analytics, has the potential to extract valuable information and hidden patterns from these large datasets [34]. Machine learning algorithms and artificial intelligence (AI) have the capability to process and analyze vast amounts of data in an efficient and accurate manner. They can uncover hidden correlations and patterns that may not be readily apparent to humans. This allows for the discovery of new relationships between biomarkers and diseases, as well as the identification of patterns in response to specific treatments. In the context of biomarkers, the application of big data and AI can significantly enhance predictive and diagnostic capabilities. Machine learning models can analyze multiple biomarkers simultaneously and generate predictive profiles that help identify diseases at early stages or predict the likelihood of a positive or negative response to a specific treatment. For example, in the field of cancer, AI algorithms can analyze genomic and molecular data from patients to identify patterns that indicate tumor aggressiveness or the probability of recurrence [40].

Furthermore, AI can also help improve the accuracy and efficiency in the interpretation of biomarkers. AI algorithms have the ability to analyze medical images, including CT scans and MRI, to identify distinctive characteristics that could be linked to particular diseases or prognoses. This can assist doctors in making more informed and precise decisions regarding diagnosis and treatment [34,35,41].

### 3.2. Types of Biomarkers

Biomarkers are any measurable characteristic that indicates the presence or absence of a disease or the biological response to a stimulus, and they are used to assess health status, diagnose diseases, track disease progression, and evaluate treatment response and other aspects related to health [42,43]. The FDA-NIH Biomarker Working Group has established seven categories to classify biomarkers according to their clinical application: susceptibility and risk, diagnosis, monitoring, prognosis, predictability, pharmacodynamics, and treatment response and safety. In addition to this clinical classification, biomarkers can also be classified based on their bodily source and type of measurement [44,45]. Here are some common types of classification for biomarkers [44,45,46,47]:

According to their biochemical nature:Genetic biomarkers: include variations in DNA, such as genetic mutations or single nucleotide polymorphisms (SNPs).Protein biomarkers: refer to specific proteins or protein profiles that can be measured in biological samples, such as blood or urine.Lipid biomarkers: are related to the lipids or fats present in the body and may be associated with cardiovascular or metabolic diseases.Metabolic biomarkers: refer to products or intermediates of metabolism that can be measured in biological samples.

According to their clinical application:Diagnostic biomarkers: used to confirm or rule out the presence of a specific disease or medical condition.Prognostic biomarkers: used to predict the progression of a disease or response to treatment.Treatment response biomarkers: used to evaluate the effectiveness of a specific treatment and adjust its dosage or duration.Disease progression biomarkers: used to monitor the advancement of a disease and evaluate its severity.Predictive biomarkers: used to predict the likelihood of a patient’s response to a specific treatment before it is administered. These biomarkers are based on biological characteristics or signals that indicate the probability of a patient responding favorably to a particular treatment.

According to their location:Blood biomarkers: found in the blood and easily accessible through blood analysis.Urinary biomarkers: found in urine and used to assess renal function and detect urinary tract diseases.Tissue biomarkers: found in specific tissues and may require biopsies or imaging studies for their retrieval.

### 3.3. NSCLC Cancer Biomarkers

Biomarkers are of utmost importance in lung cancer, serving critical roles in the diagnosis, prognosis stratification, and selection of treatment options. Given the varied nature and complexity of this disease, biomarkers are instrumental in identifying distinct subtypes of lung cancer, guiding the selection of targeted therapies, and assessing the effectiveness of treatments [48,49].

One of the extensively studied and utilized biomarkers in lung cancer is EGFR. Mutations in the EGFR gene are prevalent in specific subtypes of NSCLC and can serve as predictors for the response to EGFR tyrosine kinase inhibitors such as erlotinib or gefitinib. Molecular tests, such as polymerase chain reaction (PCR) or genetic sequencing, are commonly employed to detect these mutations. Another significant biomarker is the ALK gene (anaplastic lymphoma kinase). ALK gene fusions are observed in a subset of non-small cell lung cancers and are associated with a heightened response to ALK inhibitors such as crizotinib, ceritinib, and alectinib. Molecular testing is also used to detect ALK gene fusions, similar to EGFR [9,14]. Moreover, to these biomarkers, other biomarkers are being investigated in NSCLC, such as mutations in the KRAS gene, fusion of the RET gene, and expression of ROS1 and BRAF, among others. These biomarkers can provide additional information about the molecular characteristics and treatment response in NSCLC [27,50,51].

Besides genetic biomarkers, protein biomarkers also hold substantial importance in lung cancer. One such example is the assessment of programmed death-ligand 1 (PD-L1) expression in tumor cells, which serves as a biomarker to identify patients who might respond favorably to PD-1/PD-L1 inhibitor immunotherapy. In addition to biomarkers employed for treatment guidance, ongoing research focuses on identifying biomarkers for early diagnosis and detecting cancer recurrence in lung cancer patients. For instance, analysis of microRNAs (small non-coding ribonucleic acids) in tissue or bodily fluid samples can provide information about the presence and stage of lung cancer [32,52].

Diagnostic biomarkers by immunohistochemistry. These protein biomarkers, such as TTF-1 (Thyroid Transcription Factor-1) and p40, are directly analyzed in tissue biopsies and exhibit high specificity. TTF-1, encoded by the NKX2-1 gene, is involved in regulating the transcription of specific genes in the thyroid, lung, and diencephalon. Its presence provides valuable information about the origin of the tumor. On the other hand, p40 is a protein predominantly expressed in squamous cell carcinoma of the lung. When co-expressed, it allows for the differentiation between squamous cell carcinoma and adenocarcinoma, particularly in cases where there is poor differentiation in the initial pathological diagnosis [53]. A slightly less specific marker is Napsin A, which is a protein encoded by the NAPSA gene and is useful for distinguishing adenocarcinoma from other lung cancers. For example, a tumor that is negative for TTF-1/p40 is unlikely to be a NSCLC [42]. 

Circulating Biomarkers. They are easy to measure, but studies have not shown high specificity, but rather sensitivity, and this has increased with the discovery of new protein and miRNA-type biomarkers (non-coding RNA, single-stranded molecules of 20 to 25 nucleotides in length, which can alter gene expression after transcription and play an important role in various biological functions and are highly stable). The latter have high potential as new specific and easily measurable biomarkers in liquid biopsy [54]. Among these, miR-205 and miR-375 have been identified as specific for squamous cell carcinoma and adenocarcinoma, respectively, and miR-93, miR-221, and miR-100 have been identified as potential biomarkers in adenocarcinoma [55]. A review has shown that miR-106a, miR-125a-5p, miR-129-3p, miR-205, miR-21, miR-29b, miR-375, and miR-7 may contribute to the differentiation between small cell carcinoma and non-small cell carcinoma. There is also clear and specific evidence for miRNAs in NSCLC, such as miR-19b-3p, which is related to the regulation of the HOXA9 gene (down-regulation), implicated in a negative prognosis [55], and miR-199a-5p, which is related to the regulation of the SLC2A1 gene (down-regulation) [56].

Predictive Biomarkers. These biomarkers are primarily associated with treatment considerations. Prominent organizations such as the College of American Pathologists (CAP), the International Association for the Study of Lung Cancer (IASLC), the Association for Molecular Pathology (AMP), the American Society of Clinical Oncology (ASCO), the European Society for Medical Oncology (ESMO), and the National Comprehensive Cancer Network (NCCN) have issued recommendations regarding the predictive value of biomarkers for patient monitoring. According to their guidelines, a minimum panel of biomarkers should be included, comprising EGFR, ROS1, ALK, BRAF, NTRK, MET, and RET, to assess the suitability of targeted therapies [55,57,58].

Predictive Biomarkers for Immunotherapy. Due to the evasion of the immune system by cancer cells, one of the primary mechanisms involves immune inhibitory pathways, particularly immune checkpoint proteins present on immune cells’ surface. An example of such a protein is PD-L1 (Programmed Death Ligand 1) [52], which acts as a receptor on macrophages, antigen-presenting cells, B cells, T lymphocytes, and tumor cells, particularly in lung cancer. The upregulation of PD-L1 receptor expression deceives the immune system, leading to the inhibition of the inflammatory response. Consequently, PD-L1 serves as a preferred biomarker for immunotherapy treatment [50].

Biomarkers for Targeted Therapy. Mutations affecting the KRAS gene, particularly G12C, are implicated in the proliferation of cells and the inhibition of apoptosis in NSCLC, primarily in adenocarcinoma [59]. Another biomarker of interest is Fibroblast Growth Factor Receptor 1 (FGFR1), a tyrosine kinase that regulates various cellular processes such as proliferation, differentiation, migration, and survival. Mutation in FGFR1 contributes to the promotion of mesenchymal-predominant NSCLC [60]. Additionally, Discoidin domain Receptor 2 (DDR2), a tyrosine kinase, is considered as a biomarker, as its mutation promotes tumor growth, particularly in 30% of squamous NSCLC [61]. Furthermore, the human epidermal growth factor receptor 2 (HER2), involved in activating cell proliferation pathways such as PI3K-AKT and MEK-ERK, is found in 10 to 30% of NSCLC cases, especially in adenocarcinoma [62].

Exosomal PD-L1 has emerged as a significant biomarker in the field of NSCLC immunotherapy monitoring. It has proven to be instrumental in advancing the effectiveness of immunotherapy by serving as an indicator of the tumor cells’ adaptive response to T cells. By analyzing exosomal PD-L1, clinicians can stratify patients into responders and non-responders, aiding in treatment decision-making and prognosis assessment [63]. N6-methyladenine (m6A) is the main epigenetic RNA modification in eukaryotes, and there is significant evidence that m6A-associated proteins act as oncogenes and tumor suppressors, being abnormally expressed in NSCLC cell lines and tissues [64]. SFTA2, whose down-regulation indicates a poor prognosis in NSCLC patients, is related to increased inflammatory cells in some parts of the EMT, tumor heterogeneity, treatment efficiency, and the immune microenvironment of NSCLC [65]. Table 2 provides information on various biomarkers analyzed in NSCLC.

### 3.4. Novel NSCLC Cancer Biomarkers

The identification of new biomarkers in the field of NSCLC has revolutionized the classification of the disease, leading to more accurate and personalized treatment strategies. In the past, NSCLC classification relied primarily on histological subtypes such as adenocarcinoma, squamous cell carcinoma, and large cell carcinoma. However, this approach failed to capture the extensive heterogeneity and molecular diversity observed in NSCLC [66,67].

The advent of genomic technologies and high-throughput screening techniques has enabled the identification of specific genetic mutations, gene expression patterns, and protein markers that are associated with NSCLC. These biomarkers provide insights into the underlying molecular mechanisms driving tumor growth, metastasis, and treatment resistance. They also hold promise for predicting patient outcomes, guiding treatment decisions, and developing novel therapies. Furthermore, biomarkers associated with drug resistance, tumor recurrence, and metastasis are being actively investigated. Understanding the mechanisms underlying treatment resistance and disease progression is crucial for developing effective therapies and improving patient outcomes. Biomarkers such as TP53 mutations, VEGF expression, and TUBB3 levels, as mentioned earlier, have been linked to increased resistance to therapy, poor prognosis, and aggressive tumor behavior in NSCLC [66,67,68,69,70].

The incorporation of these biomarkers into clinical practice has the potential to revolutionize the management of NSCLC. They offer valuable insights into early detection, risk assessment, treatment selection, and treatment monitoring. By identifying patients who are likely to benefit from specific therapies or require alternative approaches, biomarkers enable a personalized and targeted approach to NSCLC management. It is important to acknowledge that the discovery and validation of biomarkers is an ongoing process. Rigorous research studies and clinical trials are necessary to establish their clinical usefulness, reliability, and reproducibility. Furthermore, the development of standardized testing methods and guidelines for biomarker evaluation is vital to ensure accurate and consistent outcomes across different healthcare settings and laboratories [67,71].

According to the analysis of Šutić et al. (2021), a group of predictive biomarkers is being studied, regarding the alteration of the KRAS gene, specifically the G12C mutation. Different drugs such as AMG510 (Sotorasib) and MRTX849 (Adagrasib) are being evaluated in patients with previously treated or metastatic NSCLC. These clinical trials, known as CodeBreak200 and KRYSTAL-12, aim to compare the effectiveness of these drugs with standard chemotherapy, such as Docetaxel, in patients with NSCLC. Another promising approach is the combination of selumetinib with docetaxel for patients with specific KRAS mutations in codons 12 or 13. The SELECT-1 trial is investigating the efficacy of this combination compared to placebo plus docetaxel in patients with locally advanced or metastatic NSCLC. Abemaciclib (LY2835219) is another drug currently being evaluated in the JUNIPER clinical trial. This study focuses on stage IV NSCLC patients who have experienced disease progression after receiving platinum-based chemotherapy and aims to compare the effectiveness of abemaciclib with erlotinib. Moreover, combinations of drugs are being explored in patients with specific genetic alterations. For instance, the NCT02743923 trial is comparing the efficacy of carboplatin, paclitaxel, and bevacizumab with cisplatin and pemetrexed in stage IIIB or IV NSCLC patients who are eligible for platinum-based chemotherapy and have not undergone prior chemotherapy. The amplification of the FGFR1 gene is also being investigated, and the use of dovitinib is being studied in previously treated advanced squamous cell NSCLC patients in the NCT01861197 trial, specifically targeting this subgroup of patients. Additionally, several ongoing clinical trials are evaluating the use of different drugs in patients with mutations or overexpression of the HER2 gene. For example, the NICHE trial is investigating the effectiveness of afatinib in previously treated advanced NSCLC patients. Another trial, DESTINYLung01, is assessing the use of trastuzumab deruxtecan (DS-8201a) in patients with unresectable and/or metastatic NSCLC with HER2 mutations or overexpression [71].

Prognostic biomarkers play a crucial role in predicting the outcome of patients with various diseases, including NSCLC. Table 3 provides a summary of some novel prognostic biomarkers and their associated outcomes [71].

TP53: The TP53 gene, which acts as a critical tumor suppressor gene, has been the subject of extensive research in different types of cancers, including NSCLC. Mutations in the TP53 gene have been found to be associated with heightened resistance to therapy in patients with NSCLC. Additionally, this gene plays a crucial role in regulating cellular growth, DNA repair processes, and programmed cell death (apoptosis). Mutations in this gene can disrupt these functions, leading to uncontrolled cell growth and impaired response to standard therapies. NSCLC patients with TP53 mutations may have a higher likelihood of treatment resistance and poorer treatment outcomes [68].

VEGF is a protein that plays a role in the process of angiogenesis, which is the formation of new blood vessels. In the context of NSCLC, elevated levels of VEGF have been linked to a less favorable prognosis. Elevated VEGF expression promotes the growth of blood vessels that supply tumors with essential nutrients and oxygen. This process facilitates tumor growth, metastasis, and recurrence. Measurement of this protein levels can provide insights into the aggressiveness of the disease and help guide treatment decisions [72].

TUBB3 (Class III Beta-Tubulin): TUBB3 is a protein involved in the organization and function of microtubules, which are essential components of the cell’s structural framework and which are crucial for cell division and intracellular transport. Increased expression of this protein in NSCLC tumors has been linked to a poor prognosis. Elevated levels of TUBB3 have been correlated with a more aggressive tumor phenotype and resistance to specific chemotherapy drugs. As a result, TUBB3 holds promise as a potential biomarker for identifying patients who may require alternative treatment approaches or targeted therapies [69].

KIAA1522: Elevated expression of the KIAA1522 gene has been correlated with a poorer prognosis and lower response rates in NSCLC. KIAA1522, a gene involved in multiple cellular processes such as cell growth and proliferation, has been associated with a more aggressive tumor behavior and diminished responsiveness to therapy. Higher expression levels of KIAA1522 may serve as an indicator of a tumor’s aggressive nature and decreased sensitivity to treatment options. Monitoring this gene levels can aid in predicting treatment response and guiding personalized treatment approaches [73].

nm23-H1: The presence of nm23-H1 has prognostic significance in NSCLC. The expression of this gene is responsible for regulating cell motility and the ability of cancer cells to spread and form metastases. When this biomarker is expressed at low levels in NSCLC patients, it has been associated with a less favorable prognosis. This suggests a higher risk of disease progression and the potential for metastasis to occur. Assessing nm23-H1 expression can help in determining the aggressiveness of the disease and informing treatment decisions [74].

## 4. Methodology for Discovering Novel Biomarkers 

### 4.1. Artificial Intelligence Algorithms

Artificial intelligence algorithms have emerged as promising tools in the prediction of biomarkers in NSCLC. These algorithms rely on the ability to process large amounts of clinical, genetic, and molecular data to identify hidden patterns and relationships that may be relevant for biomarker prediction [75,76,77]. For example, the description of disease subgroups based on genomic and clinical data allows for accurate stratification through the application of machine learning tools and algorithms to the datasets, representing a revolution in disease stratification [78]. The identification of overall survival or relapse will require a robust and accurate predictive model, which in turn requires a large volume of complex data that need to be analyzed through machine learning algorithms to detect patterns and relationships among variables and generate predictive models that contribute to the outcome, such as susceptibility to a disease such NSCLC, its recurrence, and/or morbidity-mortality [76,79,80].

Machine learning models have been used for just over 30 years in cancer detection and diagnosis, primarily artificial neural networks and decision trees. In the past decade, they have also been employed in cancer prediction and prognosis (cancer susceptibility, recurrence, and survival). Some of the frequently employed supervised algorithms include logistic regression, decision trees, support vector machines, random forest, naive Bayes, k-nearest neighbors, and neural networks. On the other hand, unsupervised algorithms such as K-means, Principal Component Analysis (PCA), and nonlinear dimensionality reduction algorithms such as t-SNE (t-distributed stochastic neighbor embedding) and UMAP (Uniform Manifold approximation and projection) have been widely utilized [78,80,81,82] (see Table 4).

The application of artificial intelligence algorithms in NSCLC involves several stages. Firstly, clinical and molecular data from patients are collected, such as demographic information, medical histories, laboratory test results, and genetic profiles. These data are used to construct datasets that contain relevant features for biomarker prediction. Next, a suitable artificial intelligence algorithm is selected for the data analysis. This may include supervised machine learning algorithms such as logistic regression, support vector machines (SVM), or neural networks, which can learn from labeled data and make predictions based on those labels. Unsupervised learning algorithms such as clustering or dimensionality reduction can also be employed to discover patterns and clusters in the data without the need for prior labels. Once the algorithm is selected, it is trained using the collected dataset. During training, the algorithm analyzes the data and adjusts its parameters to find patterns and relationships that enable the prediction of the target biomarkers. This process may involve optimizing the algorithm’s hyperparameters and cross-validation to evaluate its performance on independent datasets. Once the algorithm is trained, it is used to make predictions on new data. This may involve evaluating new patients’ genetic profiles to predict the presence or absence of specific biomarkers, estimating the risk of disease progression or treatment response, or classifying patients into subgroups based on gene expression [76,79,80].

Machine learning algorithms have evolved according to the quantity and characteristics of the data and the objective being pursued, aiming for improved classification, prediction, and visualization. There are numerous publications in biological and life sciences that include one or multiple algorithms in their methodology and results, indicating an increasing possibility of obtaining easily interpretable predictive outcomes for individual and population health [90].

### 4.2. Bioinformatics for Biomarkers Prediction

Bioinformatics plays a critical role in the identification and discovery of novel biomarkers by facilitating the efficient analysis and interpretation of vast amounts of biological data. It enables researchers to navigate through various phases of biomarker development, starting from the initial discovery phase, followed by analytical validation, evaluation of clinical utility, and ultimately leading to the clinical application of biomarkers (as shown in Figure 2). In the biomarker discovery phase, biomaterials obtained from cell assays, animal models, and clinical trials are useful, and their information (genome, transcriptome, epigenome, proteome, metabolome, and microbiome) is stored in databases such as the Gene Expression Omnibus (GEO). Statistical and bioinformatic methods are developed using this information to identify candidate biomarkers for further analytical validation using samples from cancer patients. In the validation phase, in vitro or in vivo molecular tests such as qPCR or RT-PCR and immunoassays are performed to detect the biomarker. Additionally, statistical significance tests are conducted to validate and ensure reproducibility in its detection. In the clinical evaluation phase, randomized clinical trials are conducted to test whether these biomarkers have diagnostic, prognostic, or predictive utility. Once the biomarker’s utility is demonstrated, it proceeds to the clinical use phase where it is commercialized with its respective protocol and complies with the corresponding regulatory processes [49,94,95,96].

In the biomarker discovery phase, bioinformatics methods have gained relevance due to their ability to identify differentially expressed genes, proteins, microRNAs, and peptides in healthy and affected tissues, with the aim of characterizing which of these expressed genes can potentially be a cancer biomarker. The storage of information in databases such as GEO has facilitated computational bioprospecting for the discovery of new biomarkers. Within the information provided by the GEO database, it includes the type of data, number of samples, biological type, cell line type, and whether or not they received treatment. Prior to data analysis, the data must be preprocessed by removing duplicate data and normalizing using packages such as LIMMA in the R software [66,67,95,97]. In the discovery of biomarkers, not only the analysis of differential gene expression has been important but also the use of clustering algorithms such as K-means, co-expression analysis, and investigation of central genes using tools such as Cytoscape (https://cytoscape.org/ accessed on 20 March 2023) or STRING (https://string-db.org/ accessed on 20 March 2023), gene ontology analysis (DAVID: https://david.ncifcrf.gov accessed on 20 March 2023), KEGG (https://www.genome.jp/kegg/pathway.html accessed on 20 March 2023), Gene Ontology (GO: http://geneontology.org/ accessed on 20 March 2023), and detection of associated metabolic pathways and co-expression/abundance of genes. The validation of candidate biomarker genes is performed using databases such as Gene Expression Profiling Interactive Analysis or GEPIA (http://gepia.cancer-pku.cn/ accessed on 20 March 2023), The Cancer Genome Atlas or TCGA (https://www.cancer.gov/about-nci/organization/ccg/research/structural-genomics/tcga accessed on 20 March 2023), and The Human Protein Atlas or HPA (https://www.proteinatlas.org/ accessed on 20 March 2023) [94,98,99]. The evaluation of biomarkers for cancer diagnosis can also be achieved by creating a prediction model based on the Support Vector Machine (SVM) classification algorithm, which creates a hyperplane that differentiates between two classes (presence and absence of the tumor) using two datasets, the training dataset and the test dataset. Survival analysis using Kaplan–Meier is often performed to assess the prognostic capacity of each biomarker gene and distinguish between low and high expression groups [99,100].

Below is a general methodology for discovering new biomarkers using bioinformatics tools [79,101,102]:Genomic data identification: The first step is to gather relevant genomic data for the study. This may include publicly available datasets from genomic databases, such as GenBank or the Human Genome Project Repository, or internally generated data from experiments.Data preprocessing: Once the genomic data are obtained, they need to undergo preprocessing to ensure data quality and prepare the data for analysis. This may involve data cleaning, normalization, and the removal of irrelevant or noisy data.Gene expression analysis: One of the most common ways to discover biomarkers is through gene expression analysis. This involves comparing gene expression levels between different sample groups, such as samples from patients with and without a particular disease. Techniques such as microarrays or RNA sequencing can be used to measure gene expression.Genetic variant analysis: Another strategy is to analyze genetic variants, such as mutations or polymorphisms, and their association with a specific condition. This can be achieved through the analysis of DNA sequencing data, where differences in genetic sequences among different sample groups are sought.Data mining and statistical analysis: Once the preprocessed data are available, data mining and statistical analysis techniques can be applied to identify significant patterns and associations. This may include correlation analysis, enrichment analysis of biological pathways, gene interaction network analysis, and classification or clustering analysis.Experimental validation: Biomarkers identified through bioinformatic analysis need to be experimentally validated to confirm their clinical relevance. This may involve the use of molecular biology techniques such as real-time PCR, Western blotting, or immunohistochemistry to verify the expression of the biomarkers in additional samples.Clinical application: Once validated, the new biomarkers can be used in clinical studies to assess their utility in the diagnosis, prognosis, or treatment response of a specific disease. They can also be valuable for developing personalized therapies based on the presence or absence of certain biomarkers.

The following are the bioinformatics methodologies addressed in each research on biomarker discovery for various types of cancer (Table 5).

The study by Wang et al. (2022), using the methodology presented above, discovered the genes ABCA8, ADAMTS8, ASPA, CEP55, FHL1, PYCR1, RAMP3, and TPX2 as potential diagnostic biomarkers for NSCLC. Therefore, these genes can be considered as predictive biomarkers that may help predict the presence or risk of NSCLC. However, further studies are needed to determine if these genes can also be useful as prognostic biomarkers to predict the progression or treatment outcome of NSCLC [79].

On the other hand, the study of Lai et al. (2023) to identify key genes and potential biomarkers for NSCLC used integrated microarray datasets and various bioinformatics analyses. Four microarray datasets were combined, and differentially expressed genes (DEGs) were identified. Functional clustering and pathway enrichment analyses were performed, revealing enrichment in processes such as mitotic nuclear division and cell cycle regulation. Protein–protein interaction (PPI) network analysis identified central node genes, and their potential prognostic value was confirmed through survival analysis. The markers predicted in this study are listed below [101]:ANLN: ANLN is observed to be overexpressed in multiple tumor types, including pancreatic, brain, breast, and lung cancers. It is involved in cell proliferation, and its inhibition can impede cancer cell division, migration, and invasion. Overexpression of ANLN has been associated with lung adenocarcinoma metastasis, making it a potential target for cancer therapy.CDKN3: CDKN3 exhibits overexpression in glioma and cervical cancer and is linked to poorer survival outcomes. Its expression levels fluctuate during the cell cycle, peaking during mitosis. High levels of mitotic CDKN3 expression are often observed in various human cancers.CCNB1 and CCNB2: These genes play essential roles in meiotic resumption and have been implicated in tumor cell division, proliferation, and tumor growth in several cancer types, including colorectal, pancreatic, breast, hepatocellular carcinoma, and NSCLC.KIF4A: KIF4A is involved in DNA replication and repair processes and promotes cell proliferation. It is associated with tumor size in oral carcinoma and has potential prognostic value in various solid tumors.KIF11 and MELK: Both KIF11 and MELK have been identified as oncogenes in multiple cancers and are being investigated as potential targets for cancer treatment in ongoing phase I/II clinical trials.CEP55: CEP55 is considered a promising cancer vaccine candidate and serves as a marker for predicting cancer invasion risk, metastasis, and therapeutic response.HMMR: HMMR is a microtubule-associated protein that regulates mitosis and meiosis. Aberrant expression of HMMR disrupts the cell division process and is associated with cancer risk and progression across various tumor types.ASPM: ASPM has emerged as a predictor of tumor aggressiveness and prognosis in bladder, prostate, and endometrial cancers.CENPF: CENPF serves as a proliferative marker for malignant tumor cell growth.BUB1: BUB1 is a serine/threonine-protein kinase that plays a critical role in oncogenesis, chromosome arrangement, and spindle assembly.

## 5. Challenges and Future Perspectives

In the field of cancer biomarkers, a key challenge is the need for a predefined panel of diagnostic markers that can be efficiently tested on small tumor biopsies with a quick turnaround time. Additionally, the emergence of new biomarkers and limited awareness of available testing options contribute to gaps in testing. Expanding the testing to include a broader range of biomarkers is crucial for personalized treatment options and access to clinical trials. Another challenge lies in the standardization of ctDNA analysis, a minimally invasive approach for tumor profiling. While ctDNA analysis offers advantages such as easier blood sampling and faster results, there are issues regarding standardization, sensitivity, and technical aspects that need to be addressed, including the detection of gene fusions in plasma and cost-effectiveness of the testing [115].

The current trend in lung cancer biomarker detection aims to achieve more accurate results through less invasive procedures. However, this approach presents new challenges for pathologists. In practical scenarios, it can be difficult to combine all the necessary techniques when working with the biological samples received in the laboratory. The analysis of phenotypic and genotypic characteristics in small tissue specimens becomes particularly complex due to the tumor’s heterogeneity, especially when using multiplex immunohistochemistry (IHC) or next-generation sequencing (NGS) approaches that involve comprehensive panels. Tumor heterogeneity is a dynamic phenomenon that can complicate the assessment of certain biomarkers during the follow-up of lung cancer patients. While automated tests offer the advantage of delivering rapid results, the cost associated with performing certain analyses systematically, such as NGS, is significant compared to the level of care provided based on the obtained results [116].

Ensuring access to novel diagnostic tools for all patients is another significant challenge. Barriers such as limited testing capabilities, geographic limitations, and guidelines hinder the availability of accurate and timely molecular diagnoses. Overcoming these barriers requires regional coordination, collaboration among accredited laboratories, and consideration of factors like turnaround time and cost-effectiveness. Reimbursement and financial issues must also be resolved to support the adoption of innovative diagnostic techniques. Lastly, the harmonization of molecular diagnostics in NSCLC involves collaboration and consensus among various stakeholders, including physicians, pathologists, molecular laboratories, and healthcare institutions. Achieving consensus on standard diagnostic tools and approaches, addressing performance variations, and involving the pharmaceutical and diagnostics industry through education and pilot projects are crucial for harmonization. This long-term process requires evidence-based innovation and shared responsibility to improve the implementation of biomarker testing and enhance patient outcomes in cancer management [115].

The adoption of AI in healthcare faces several challenges. The lack of transparency in how deep learning systems generate predictions, especially weakly supervised models, is a major concern. Techniques such as attention weights and knowledge distillation offer potential solutions by providing insights into the network’s decision-making process. Ensuring the generalizability of AI systems to specific data types is another challenge, as models may exhibit inflated performance for low-frequency events and unintended biases can arise from data generation. Addressing these issues requires careful assessment of predictions and consideration of potential sources of noise. The requirement for labeled data is also a hurdle, but crowdsourcing annotations and utilizing self-supervised or unsupervised learning techniques can alleviate the labeling burden. The specificity of domain knowledge in healthcare further complicates AI adoption, necessitating the development of networks that mimic medical practitioners’ expertise and incorporating multi-modal data for more accurate predictions. Overcoming these challenges will contribute to the successful integration of AI in healthcare, leading to improved clinical outcomes [117].

## 6. Conclusions

NSCLC is a significant public health concern with high mortality rates. Advances in genomic data and bioinformatics tools have improved the possibilities for early diagnosis, effective treatment, and follow-up through the use of biomarkers. Biomarkers combined with clinical data can predict outcomes and guide specific treatments. However, the impact of new biomarkers on mortality and treatment efficacy in NSCLC is challenging due to limited evidence. The identification and application of relevant biomarkers, both new and existing, will shape the future of precision medicine in NSCLC.

Precision medicine, which focuses on personalized and targeted treatments based on individual genetic and molecular characteristics, has transformed the field of healthcare. It has led to the development of targeted therapies and improved treatment outcomes in oncology. Biomarkers play a crucial role in precision medicine by providing measurable indicators of disease characteristics and helping tailor treatment approaches. The integration of big data and AI in biomarker analysis and interpretation further enhances the potential for personalized medicine. However, challenges remain in terms of data analysis, interpretation, and the adoption of precision medicine approaches in clinical practice. Continued research and advancements in biomarker discovery and utilization are essential for further improving patient outcomes in lung cancer and other diseases.

## Figures and Tables

**Figure 1 cancers-15-03474-f001:**
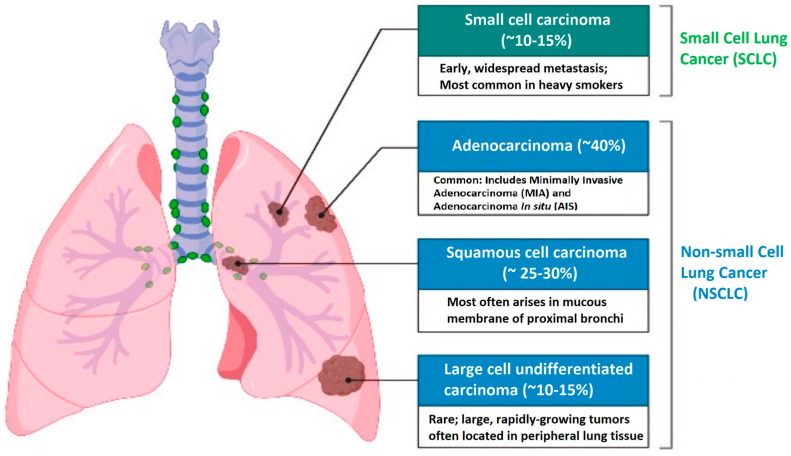
Histological classification of lung cancer.

**Figure 2 cancers-15-03474-f002:**
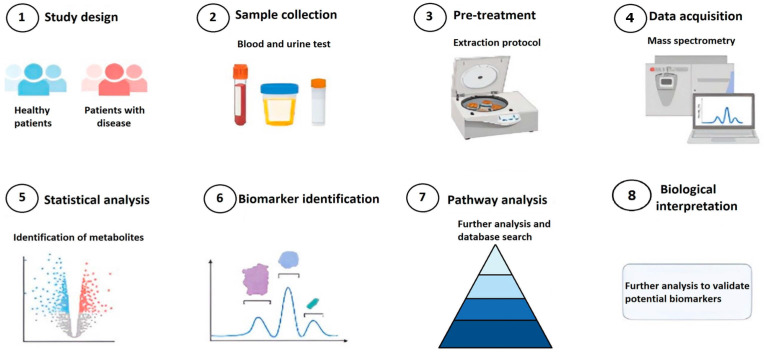
Methodology for biomarker identification.

**Table 1 cancers-15-03474-t001:** Advantages and disadvantages of different diagnostic methods for NSCLC (MRI—Magnetic Resonance Imaging, PET—Positron Emission Tomography, CT—Computed Tomography, FISH—Fluorescence In Situ Hybridization, PCR—Polymerase Chain Reaction, IHC—Immunohistochemistry, FDG—Fludeoxyglucose F18).

Diagnostic Method	Advantages	Disadvantages
MRI	No ionizing radiation exposure	Limited availability and restricted access
	Detailed imaging of soft tissues	Lower sensitivity for detecting small lesions
PET	Detects metabolic and molecular changes	Higher cost and limited availability
	High sensitivity for detecting metastasis	Potential for false positives due to FDG accumulation
CT	Widely available and rapid access	Exposes the patient to ionizing radiation
	High spatial resolution and early tumor detection	Potential for false positives due to benign lesions
FISH Biomarkers	Provides genetic information about specific cancer subtypes	Requires specialized laboratory analysis
PCR Biomarkers	Provides genetic information about specific cancer subtypes,	Requires specialized laboratory analysis
IHC Biomarkers	Provides protein expression information, Helps differentiate cancer subtypes	Requires specialized personnel and equipment, results may vary depending on the method used
Next Generation Sequencing Biomarkers	Provides comprehensive genetic information	Requires specialized laboratory analysis
Liquid Biopsy	Non-invasive and lower risk for the patient	Lower sensitivity compared to tissue biopsy
	Enables monitoring of genetic changes over time	Potential for false negatives due to low concentration
Tissue Biopsy	Provides tissue samples for histopathological analysis	Invasive procedure with associated risks
	High precision and detection of genetic mutations	Potential complications such as bleeding or infection

**Table 2 cancers-15-03474-t002:** Biomarkers in NSCLC.

Biomarkers in Nsclc					
Diagnostic Biomarkers in NSCLC					
Immunohistochemistry			Circulating tumor proteins		
TTF-1 (Thyroid Transcription Factor-1)			Cytokeratin 19 fragment (CYFRA 21-1)		
p40			Carcinoembryonic Antigen (CEA)		
Napsin A			Squamous cell carcinoma antigen (SCCA)		
			Carbohydrate antigen 125 (CA125)		
**microRNA (miRNA)**					
miR-205	miR-106a	miR-29b			
miR-375	miR-125a-5p	miR-375			
miR-93	miR-129-3p	miR-7			
miR-221	miR-205	miR-19b-3p			
miR-100	miR-21	miR-199a-5p			
**Predictive Biomarkers in NSCLC**					
Targeted Therapy		For Inmunotherapy		Novel predictive biomarkers	
Biomarker	Therapy	Biomarker	Antibody	Targered therapy	Inmunotherapy
EGFR	Afatinib	PD-1	Atezolizumab	KRAS	Exosome PD-L1
	Erlotinib		Pembrolizumab	FGFR1	m6A methylation
	Gefitinib		Durvalumab	DDR2	SFTA2
	Osimertinib		Nivolumab	HER2	TIL’s
					TIM-3
ROS1	Entrectinib				TMB
	Ceritinib				
	Crizotinib				
ALK	Alectinib				
	Crizotinib				
	Lorlatinib				
MET	Tepotinib				
	Capmatinib				
RET	Selpercatinib				
	Praseltinib				
BRAF	Trametinib				
	Dabrafenib+				
NTRK (1,2,3)	Larotrectinib				

**Table 3 cancers-15-03474-t003:** Novel prognostic biomarkers for NSCLC.

Biomarker	Outcome
TP53	Resistance to Therapy increased.
VEGF	Poor prognosis, metastasis, tumor recurrence.
TUBB3	Poor prognosis
KIAA1522	Poor prognosis and lower response rate
nm23-H1	Low levels poor prognosis
TGF-β	Poor prognosis
LAG-3	Better survival
NLR&PLR	Worse overall survival
Ki-67	Poor prognosis

**Table 4 cancers-15-03474-t004:** Supervised and unsupervised machine learning algorithms used in biological and life sciences.

Algorithm	Advantages	Limitations	References
Supervised Algorithms			
Logistic Regression	High power for supervised classification with a dichotomous variable	Not useful for continuous variables	Yang, 2022 [83]
Support Vector Machine	Applied in non-linear models and survival prediction in cancer and demographic studies, among others. Good control of overfitting and good classifier	Complex algorithm structure. Training is slower.	Huang, 2022 [84]
Decision Trees	Easy algorithm for data training. Used in diagnostic protocols	Can have overfitting problems, especially when there is a significant increase in branching in internal nodes	Lai, 2020 [75]; Batra, 2022 [85], 2022 [7]
Random Forest	Good predictive algorithm used in medicine in different imaging studies and recently in biomarker studies	May have overfitting problems	Batra, 2022 [85]; Handelman, 2018 [80]
Naïve Bayes	Still used in symptom characterization, complication prediction, imaging data, and demographic data.	As it is based on probabilistic statistical models, it can assume that attributes are independent. Redundant attributes can induce classification errors	Yang, 2023 [86]
K-Nearest Neighbor	Used as a classification and prediction algorithm in demographic models and genomic data, among others. Tolerant to noisy and missing data	Can assume that data attributes are equally important and may have similar classifications. Computationally complex with increasing data and attributes	Podolsky, 2016 [82]
Artificial Neural Networks	Algorithmic model capable of classifying and predicting based on a combination of parameters and applying it at the same time.	May have overfitting with too many attributes, and the optimal network structure is determined for experimentation	Lian, 2022 [87]; Civit-Masot, 2022 [88]
Unsupervised Algorithms			
K-Means	Widely used algorithm in biological and medical research and is easy to adapt and understand. Performs well on large datasets	The number of K needs to be manually assigned. Outliers can generate incorrect clusters. Scaling issues with the number of dimensions	Huang, 2021 [89]
Principal Component Analysis (PCA)	Linear dimensionality reduction algorithm that allows pattern observation and generates independent variables called principal components. Widely used in biological and genomic data observation	Does not allow non-linear dimensionality reduction. Lack of data standardization can be detrimental to results and information loss	Shin, 2018 [90]
t-SNE	Algorithm that enables visualization of high-dimensional datasets. Frequently used with PCA in biological and life sciences, primarily in omics analysis	Some issues when applied to non-linear parameter dimensionality reduction	Islam, 2021 [91]; Wang, 2021 [92]
UMAP	Next-generation algorithm that, similar to t-SNE, enables visualization of high-dimensional datasets. Offers higher accuracy when working with non-linear structures. Widely used in omics analysis	Currently limited to dimensional reduction due to its relative lack of familiarity	Islam, 2021 [91]; Nascimben, 2022 [93]

**Table 5 cancers-15-03474-t005:** Methodologies using bioinformatics for the prediction of biomarkers for different types of cancer. TCGA: The Cancer Genome Atlas, GEO: Gene Expression Omnibus, LIMMA: Linear Models for Microarray Data, STRING: Search Tool for the Retrieval of Interacting Genes/Proteins, GO: Gene Ontology, KEGG: Kyoto Encyclopedia of Genes and Genomes, FUNRICH: Functional Enrichment Analysis Tool, DAVID: Database for Annotation, Visualization, and Integrated Discovery, GEPIA: Gene Expression Profiling Interactive Analysis, Affy: Affymetrix, HMDD: Human MicroRNA Disease Database, BLAST: Basic Local Alignment Search Tool, miRDB: MicroRNA Target Prediction Database.

Author	Type of Cancer	Type of Data	Database	Data Preprocessing and Differentially Expressed Genes (DEGs)	MicroRNA Target Prediction	Protein–Protein Interaction (PPI)	Functional Enrichment Analysis	Validation
Zhang et al., 2020 [103]	Bladder	Gene	TCGA-BLCA, GEO	R software	Not Realized	CytoHubba, STRING	GO, KEGG, FUNRICH, DAVID	Oncomine database, GEPIA
Sarafidis et al., 2022 [95]	Bladder	Gene	GEO (metanalysis)	Affy, LIMMA R packages, outlier removal quality control	Not Realized	STRING, Cytoscape	GO, KEGG, Disease Ontology (DO), Reactome	GEPIA2, TCGA, Human Protein Atlas (proteomics, RNA-Seq), survival analysis
Pandi et al., 2022 [96]	Breast	Gene	GEO	R LIMMA package, GEO2R	Not Realized	STRING, Cytoscape	Enrichr, KEGG	TCGA-BRCA, GEPIA, survival analysis
Xu et al., 2022 [104]	Breast	Gene	GEO	R LIMMA package	Not Realized	STRING	GO, KEGG	GEPIA, survival analysis
Wu et al., 2022 [105]	Breast	Gene	TCGA	R LIMMA package	Not Realized	STRING	GO, clusterProfiler R package	Breast Cancer Gene-Expression Miner v4.8 (bc-GenExMiner v4.8), survival analysis
Fadaka et al., 2019 [106]	Colon	microRNA	miRBase (https://www.mirbase.org/ accessed on 20 March 2023), miR2Disease (http://www.mir2disease.org/ accessed on 20 March 2023), HMDD (http://www.cuilab.cn/hmdd accessed on 20 March 2023), y miRCancer (http://mircancer.ecu.edu/ accessed on 20 March 2023), BLAST	R software	miRDB (http://www.mirdb.org/index.html accessed on 20 March 2023), TargetScan (https://www.targetscan.org/vert_72/ accessed on 20 March 2023) y mirDIP (http://ophid.utoronto.ca/mirDIP/index.jsp accessed on 20 March 2023)	STRING, Cytoscape	DAVID, GO, KEGG	Gene correlation (gbCRC) at http://gbcrc.bioinfo-minzhao.org/ accessed on 20 March 2023
Dai et al., 2019 [107]	Colon	Gene	GEO (systematic review)	R software, BRB array tools	Not Realized	STRING, Cytoscape	FunRich (http://www.funrich.org/ accessed on 20 March 2023), KEGG, DAVID	The Human Protein Atlas (HPA), The Cancer Genome Atlas (TCGA), survival analysis
Li et al., 2021 [108]	Colon	Gene	GEO	R limma package	Not Realized	STRING, Cytoscape	KEGG, DAVID	TCGA, GEPIA, survival analysis
Hammad et al., 2021 [77]	Colon	Gene	GEO	R software	Not Realized	STRING, Cytoscape	KEGG, DAVID	GEPIA, survival analysis (PROGgene)
Paksoy and Hilal, 2022 [109]	Colon	Gene	https://figshare.com/articles/dataset/The_microarray_dataset_of_colon_cancer_in_csv_format_/13658790/1 accessed on 20 March 2023	Synthetic Minority Oversampling Technique, or SMOTE method	Not Realized	Not Realized	Not Realized	Not Realized
Wang et al., 2021 [110]	Gastric	Gene	TCGA	R limma package	Not Realized	Cytoscape	GO, KEGG	GEPIA, survival analysis, Jiangsu Province Yixing People’s Hospital
Liu et al., 2022 [111]	Gastric	Gene	GEO, TCGA	R limma package, clustering analysis (Bioconductor)	Not Realized	STRING, Cytoscape	KEGG, DAVID	TCGA, survival analysis
Lvu et al., 2020 [97]	Glioma	mRNAsi	TCGA	EdgeR method	Not Realized	Not Realized	GO, KEGG	Survival analysis, Chinese Glioma Genome Atlas (CGGA) (http://www.cgga.org.cn/ accessed on 20 March 2023)
Liao et al., 2020 [112]	Lung	mRNAsi	TCGA	R LIMMA package	Not Realized	STRING, Cytoscape	GO, KEGG, DAVID	GEPIA, survival analysis
Gong et al., 2021 [49]	NSCLC	Gene	GEO	GEO2R	Not Realized	STRING, Cytoscape	KEGG, DAVID	GEPIA, survival analysis, Oncomine database
Wu et al., 2019 [100]	Ovarian	microRNA	GEO (Systematic review and Metanalysis)	edgeR package of R	Not Realized		GO, KEGG, DAVID	Survival analysis
Chen et al., 2020 [99]	Ovarian	Gene	GEO	R software	Not Realized	GeneMANIA (https://genemania.org/ accessed on 20 March 2023)	KEGG, DAVID	Survival analysis, Dataset GSE9891
Zahra et al., 2022 [113]	Ovarian	Gene	TCGA, UK BioBank, cBioPortal	R software	Not Realized	Not Realized	Not Realized	Not Realized
Shi et al., 2022 [114]	Pancreas	Gene	GEO	GEO2R		STRING, Cytoscape	GO, KEGG, DAVID	GEPIA, Survival analysis
Yuan et al., 2017 [98]	Prostate	Gene	GEO	Affy, LIMMA R packages		STRING, Cytoscape	GO, DAVID	Protein Atlas Database, Oncomine database
Lombe et al., 2022 [94]	Prostate	microRNA	GEO	GENT2 (http://gent2.appex.kr/gent2/ accessed on 20 March 2023)	TargetScan Human, miRDB, DIANA microT	STRING, Cytoscape	GO, KEGG, DAVID	GEPIA, survival analysis
Author	Prediction of drug-gene interaction		Evaluation of pronostic biomarkers			Protein acquisition, 3D modeling and protein visualizer		In vitro Validation
Zhang et al., 2020 [103]	Not Realized		Not Realized			Not Realized		Not Realized
Sarafidis et al., 2022 [95]	Not Realized		Least Absolute Shrinkage and Selection Operator (LASSO) regression			Not Realized		Not Realized
Pandi et al., 2022 [96]	Not Realized		Not Realized			Not Realized		Not Realized
Xu et al., 2022 [104]	Not Realized		Not Realized			Not Realized		Not Realized
Wu et al., 2022 [105]	DrugBank, Cytoscape		Not Realized			Not Realized		Not Realized
Fadaka et al., 2019 [106]	Not Realized		PrognoScan (http://dna00.bio.kyutech.ac.jp/PrognoScan/ accessed on 20 March 2023)			Not Realized		Not Realized
Dai et al., 2020 [107]	Not Realized		Not Realized			Not Realized		Not Realized
Li et al., 2021 [108]	Not Realized		Not Realized			Not Realized		Not Realized
Hammad et al., 2021 [77]	Not Realized		Prediction model with Support Vector Machine (SVM classifier)			Not Realized		Not Realized
Paksoy and Hilal, 2022 [109]	Not Realized		Random Forest, Desicion Trees, Gaussian Bayes			Not Realized		Not Realized
Wang et al., 2021 [110]	Not Realized		Not Realized			Not Realized		Gastric cell lines (AGS, HGC27 and MKN45) and normal gastric mucosa cells, FISH, RNA extraction, qRT-PCR
Liu et al., 2022 [111]	Drug-Gene Interaction database (DGIdb), Cytoscape		Not Realized			Not Realized		Not Realized
Lvu et al., 2020 [97]	Not Realized		Estimation of mRNAsi using one-class logistic regressionmachine learning (OCLR), Least Absolute Shrinkage and Selection Operator (LASSO) regression			Not Realized		Not Realized
Liao et al., 2020 [112]	Not Realized		Estimation of mRNAsi using one-class logistic regressionmachine learning (OCLR)			Not Realized		Not Realized
Gong et al., 2021 [49]	Not Realized		Not Realized			Not Realized		A549 and HBE normall cells, via qPCR
Wu et al., 2019 [100]	Not Realized		Not Realized			Not Realized		Not Realized
Chen et al., 2020 [99]	Not Realized		Not Realized			Not Realized		Not Realized
Zahra et al., 2022 [113]	Not Realized		Not Realized			Uniprot, RSCB PDB (Protein Data Bank),Phyre2, Swissmodel, Alpha fold, Missense3D tool, YASARA, PYMOL, PROVEAN		Not Realized
Shi et al., 2022 [114]	Not Realized		Not Realized			Not Realized		Four PDA cell lines (AsPC-1, SW1990, PANC-1, and BxPC-3) and a normal human pancreatic ductal epithelial cell line (HPDE), qRT-CPR
Yuan et al., 2017 [98]	Not Realized		Not Realized			Not Realized		Not Realized
Lombe et al., 2022 [94]	Not Realized		Not Realized			Not Realized		MicroRNAs via qPCR

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
