# Peer review of "Advances in Genomic Data and Biomarkers: Revolutionizing NSCLC Diagnosis and Treatment"

_cancers, 2023, doi:10.3390/cancers15133474_

Round 1

Reviewer 1 Report

The manuscript "Advances in Genomic Data and Biomarkers: Revolutionizing 2 NSCLC Diagnosis and Treatment' by Restrepo et al., have discussed the advancements in genomic data, bioinformatics tools, and the utilization of biomarkers that have improved early diagnosis, effective treatment strategy, and follow-up in non-small cell lung carcinoma patients. The manuscript is very well written and very well described; however, I have some minor corrections:

1. In many place the NSCLC full-form has been used and some places the abbreviated form used. Please use the abbreviated NSCLC in those line; e.g. Line 165, 174, 178, 209, 232, 482 etc.

Author Response

Thank you for your feedback, and we have made the necessary changes.

Reviewer 2 Report

This paper was reviewer paper and covered and explained about  pathophysiology, histology and classification, Risk factors and epidemilogy of Non-small cell lung cancer and cancer biomarkers. This review paper was followed many paper about  Non-small cell lung cancer. This paper is excellent  and This paper will be useful of  Non-small cell lung cancer's research.

I have no claim about the Quality of English language .  

Author Response

(The authors gave the same response as above.)

Reviewer 3 Report

In their manuscript “Advances in Genomic Data and Biomarkers: Revolutionizing NSCLC Diagnosis and Treatment”, Juan Carlos Restrepo et al. reviewed several aspects related to biomarkers in non-small cell lung cancer (NSCLC) like the recent advancements in genomic data, bioinformatics tools, and the utilization of biomarkers in early diagnosis, effective treatment, and prognosis of the disease.

The manuscript is relevant for the field and presented in a well, structured manner. After a brief introduction about lung cancer and modern methods of diagnosis and follow-up worldwide, authors describe several aspects that characterize NSCLC (risk factors and epidemiology; pathophysiology, histology and classification; diagnosis and treatment). A special attention is paid to the mutations found in NSCLC, as well as to the advantages and disadvantages of the various diagnostic methods. Then, in section “3. Precision medicine and biomarkers” authors highlight aspects of precision medicine, types of biomarkers and NSCLC biomarkers. Further, authors detailed the methodology for discovering novel biomarkers like artificial intelligence algorithms, with their advantages and limitations, the critical role of bioinformatics for biomarkers prediction, with a list of predicted markers in NSCLC. The integration of global data and artificial intelligence (AI) further enhances the potential for personalized medicine through advanced biomarker analysis. In the end,  authors present the challenges and future perspectives. The adoption of precision medicine approaches in clinical practice posed additional challenges and emphasizes the integration of biomarkers with advanced technologies.

The review is clear, comprehensive and of relevance to the field. There are not many similar reviews published recently, and this current review is relevant and of interest to the scientific community. The iconography is appropriate, and easy to interpret and understand. Statements and conclusions are drawn coherent and supported by the listed citations. The cited references are relevant, and around 80% were published within the last 5 years; the bibliography does not include many self-citations.

Minor editing of English language required.

Author Response

(The authors gave the same response as above.)

Reviewer 4 Report

The authors presented here a general review on NSCLC biomarkers, discussing 

Table 2. Ramucirumab is NOT an EGHFR TKI, but an anti-angiogenetic agent used with erlotinib in EGFR mutated patients. Please correct.

The authors did not consider markers of IO resistance, as STK11 mutations or c-MET amplification in EGFR mutated tumors, please add a brief chapter on markers of resistance and not only positive markers 

Table 5 is difficult to consult, please change 

Good

Author Response

(The authors gave the same response as above.)

Round 2

Reviewer 4 Report

None

None